# LATENTLOGIC: Learning Logic Rules in Latent Space over Knowledge Graphs

**Junnan Liu**[1], **Qianren Mao**[2], **Chenghua Lin**[3], **Yangqiu Song**[4], **Jianxin Li**[1,2,*]

[1]School of Computer Science and Engineering, Beihang University, Beijing, P.R.China.

[2]Zhongguancun Laboratory, Beijing, P.R.China.

[3]Department of Computer Science, University of Manchester, U.K.

[4]Department of Computer Science and Engineering, HKUST, Hong Kong SAR, China.

liujn@act.buaa.edu.cn, {maoqr,lijx}@zgclab.edu.cn, chenghua.lin@manchester.ac.uk, yqsong@cse.ust.hk

## Abstract

Learning logic rules for knowledge graph reasoning is essential as such rules provide interpretable explanations for reasoning and can be generalized to different domains. However, existing methods often face challenges such as searching in a vast search space (e.g., enumeration of relational paths or multiplication of high-dimensional matrices) and inefficient optimization (e.g., techniques based on reinforcement learning or EM algorithm). To address these limitations, this paper proposes a novel framework called LATENTLOGIC to efficiently mine logic rules by controllable generation in the latent space. Specifically, to map the discrete relational paths into the latent space, we leverage a pre-trained VAE and employ a discriminator to establish an energy-based distribution. Additionally, we incorporate a sampler based on ordinary differential equations, enabling the efficient generation of logic rules in our approach. Extensive experiments on benchmark datasets demonstrate the effectiveness and efficiency of our proposed method.

## 1 Introduction

Knowledge graphs usually contain collections of real-world facts encoded in triplets with entity and relation information, and find broad applications in across multiple domains (Lukovnikov et al., 2017; Xiong et al., 2017a; Wang et al., 2018; Zhang et al., 2019; Huang et al., 2019; Tang et al., 2023a,b). Despite some knowledge graphs holding hundreds of millions of triples, they still suffer from incompleteness, whereby many valid triples are missing since it is impractical to identify them all manually. Therefore, a fundamental and essential task in knowledge graphs is to utilize existing facts to predict the missing ones.

Recent studies have focused on learning logic rules from knowledge graphs and utilizing these learned rules to predict absent facts. An example

---

* Jianxin Li is the corresponding author.

of such a rule is $\forall X, Y, Z$ nationality$(X, Y) \leftarrow$ classmate$(X, Z) \wedge$ nationality$(Z, Y)$, indicating that if $Z$ is the classmate of $X$ and has a nationality of $Y$, then $X$ is likely to have a nationality of $Y$. This rule can be applied to deduce the nationalities of new individuals. Compared to other methods such as knowledge graph embedding approaches (Bordes et al., 2013; Sun et al., 2019; Li et al., 2022), the rule-based method (Zhang et al., 2020) is more interpretable and can be applied to inductive scenarios (Teru and Hamilton, 2019).

Most rule-based methods involve enumerating relational paths as candidate rules, followed by assigning weights to each rule to indicate their quality (Lao and Cohen, 2010; Richardson and Domingos, 2006; Yang et al., 2017; Sadeghian et al., 2019). When the scale of the knowledge graph expands, these methods face the challenge of exponentially growing search space. To overcome this problem, RNNLogic (Qu et al., 2021) introduces a rule generator and a reasoning predictor to separate rule generation from rule weight learning. However, the optimization process based on the Expectation-Maximization (EM) algorithm tends to have slow convergence, leading to extended training periods. Another line of research utilizes reinforcement learning (RL) to search for logic rules by making sequential decisions (Xiong et al., 2017b; Lin et al., 2018; Das et al., 2018; Lu et al., 2022). Nevertheless, RL-based methods often encounter challenges such as large action spaces and sparse rewards during training. As a result, efficiently mining high-quality logic rules for knowledge graph reasoning remains a challenging task.

In this paper, we propose a novel framework named **LATENTLOGIC**, which overcomes the aforementioned challenges. Our approach bypasses the enumeration of relational paths by employing controllable sampling in latent space. Furthermore, each component of LATENTLOGIC is trained in an end-to-end fashion, avoiding the indirect and

inefficient optimization procedure. Concretely, we utilize a VAE-based autoencoder (Kingma and Welling, 2014; Li et al., 2020) to map discrete relational paths into a low-dimensional latent space. We employ a discriminator to measure the semantic coherence between the latent vector and the rule head, thereby creating an energy-based distribution in the latent space. To obtain latent vector samples that correspond to the desired rule head, we employ a sampler based on ordinary differential equations (Song et al., 2021; Nie et al., 2021). The latent vectors are used to generate rule bodies for the given rule head using the VAE generator.

Extensive experiments demonstrated the computational efficiency of LATENTLOGIC in comparison to previous rule learning methods (Yang et al., 2017; Sadeghian et al., 2019; Qu et al., 2021), indicating that LATENTLOGIC exhibits enhanced scalability for mining logic rules on larger-scale knowledge graphs. Furthermore, through experiments on two commonly used benchmark datasets, FB15k-237 (Toutanova and Chen, 2015) and WN18RR (Dettmers et al., 2018), we observed that LATENTLOGIC successfully generates high-quality logic rules for knowledge graph reasoning and evidently outperform the salient baseline methods.

## 2 Framework

**Problem Definition**  First, we introduce some definitions and notations. A knowledge graph $\mathcal{G} = (\mathcal{V}, \mathcal{T}, \mathcal{R})$ is usually defined by a triple set $\mathcal{T} = \{(h, r, t)\} \subseteq \mathcal{V} \times \mathcal{R} \times \mathcal{V}$, where $\mathcal{V}$ denotes an entity set, and $\mathcal{R}$ represents a relation set. In this paper, we aim to learn logic rules in the conjunctive form $\forall \{X_i\}_{i=0}^{l} : r(X_0, X_l) \leftarrow r_1(X_0, X_1) \wedge \cdots \wedge r_l(X_{l-1}, X_l)$ from the given knowledge graph. A logic rule, which can be abbreviated as $r \leftarrow r_1 \wedge \cdots \wedge r_l$, consists of a rule head, denoted as $r$, and a rule body (relational path), represented as $r_1 \wedge \cdots \wedge r_l$.

**Framework Overview**  Our approach converts rule learning problems to controllable generation problems by developing a generative model, denoted as $p_\theta(\mathcal{P}|s)$, to generate rules given a specific rule head $s$. Here, $\mathcal{P}$ represents the rule body, i.e., relational path, $\mathcal{P} = (r_1, r_2, \ldots r_l)$. The main innovation behind our approach is that we substitute the enumeration of the relational path with the sampling of latent vectors, significantly reducing training overhead. Additionally, to accomplish our

aim, we incorporate a relational path autoencoder and a rule discriminator that operates in the latent space. As shown in Fig. 1, we firstly utilize a VAE-based *Relational Path Autoencoder*, which consists of an encoder $\mathcal{E}$ and a decoder $\mathcal{D}$, to compress relational paths into a low-dimensional latent space. To perform controllable generation of rule bodies, we need to construct a joint distribution of latent vectors and rule heads for sampling. Therefore, we introduce a *Rule Discriminator* on the latent space to form the joint distribution represented by the energy-based model. This joint distribution allows us to obtain latent vectors associated with the desired rule head by using a sampler based on an ordinary differential equation. Finally, we can decode the latent vectors into rule bodies using the VAE decoder, allowing us to generate rules given certain rule heads.

### 2.1 Relational Path Autoencoder

Note that each rule body $r_1 \wedge \cdots \wedge r_l$ can be considered a sequence of relations $[r_1, \ldots, r_l]$. Such sequences of relations can be effectively modeled by sequence neural networks (Das et al., 2017; Kotnis et al., 2021; Liu et al., 2022), and thus we introduce RNN (Hochreiter and Schmidhuber, 1997) to parameterize the relational path autoencoder. Specifically, we map relational path $\mathcal{P}$ into latent vector $z$ using an RNN-based encoder $q(z|\mathcal{P})$, and an RNN-based decoder $p(\mathcal{P}|z, q)$ that maps $z$ into the relational path $q$ is a unified query embedding for all inputs. Our decoder does not use an autoregressive approach. Instead, it takes the unified query embedding $q$ and positional embedding as input, simultaneously generating relational paths. We optimize the encoder and decoder parameters for each input relational path $\mathcal{P}$ with the objective:

$$
\begin{aligned}
\mathcal{L}_{\text{VAE}}(\mathcal{P}) = & - \mathbb{E}_{q(z|\mathcal{P})}[\log p(\mathcal{P}|z, q)] \\
& + \text{KL}\left(q(z|\mathcal{P})||\mathcal{N}(0, I)\right),
\end{aligned} \tag{1}
$$

where $\text{KL}(\cdot||\cdot)$ is the Kullack-Leibler divergence that pushes $q$ to be close to the prior $\mathcal{N}(0, I)$.

### 2.2 Rule Discriminator

Now we aim to model the joint distribution $p(z, s)$, where $z$ denotes the latent vector of a relational path and $s$ represents a desired rule head. This joint distribution can be represented as $p(z, s) = p_{\text{prior}}(z)p(s|z)$, where $p_{\text{prior}}(z)$ is the prior distribution, i.e., standard Gaussian distribution, and $p(s|z)$ is conditional distribution on $s$ given $z$. Inspired by Nie et al. (2021), we define $p(s|z)$ as an

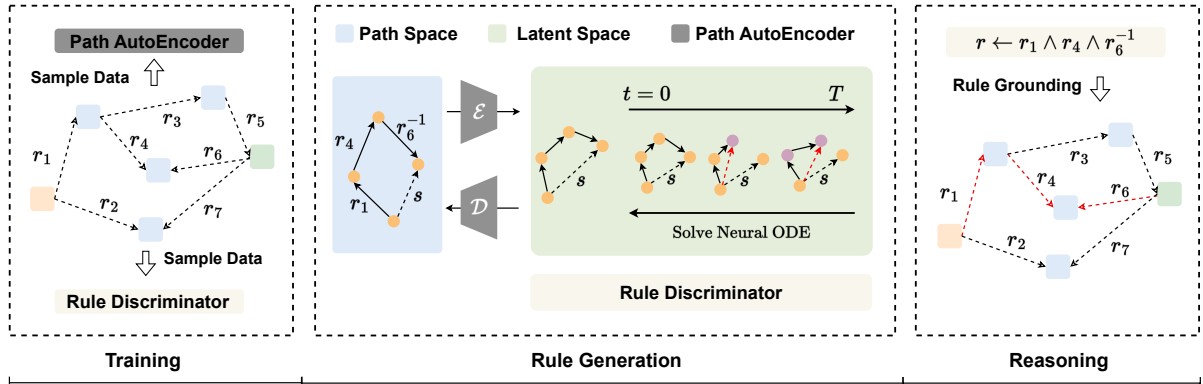

Figure 1: Framework overview of LATENTLOGIC. We first train the rule discriminator and relational path autoencoder. Then we generate logic rules by sampling on the latent space. The generated rules can be used to answer queries over knowledge graphs.

energy-based model (EBM) (LeCun et al., 2006) as follows:

$$
\begin{aligned}
p(s|\boldsymbol{z}) &\propto e^{-E_\theta(s|\boldsymbol{z})}, \\
E_\theta(s|\boldsymbol{z}) &= -f(g(\boldsymbol{z}))[s] + \text{const},
\end{aligned}
\tag{2}
$$

where $\text{const} = \log \sum_{s'} \exp(f(g(\boldsymbol{z}))[s'])$ is a normalization term corresponding to all rule heads, $g(\cdot)$ is the fixed VAE decoder that decodes $\boldsymbol{z}$ into $\mathcal{P}$, and $f$ indicates a neural network that takes $g(\boldsymbol{z})$ and $s$ as input and produces a score that measures how well $s$ is carried in $\boldsymbol{z}$. The function $f$ can be any advanced model if it measures the consistency between relational paths and the rule heads (e.g., other rule mining models like neural logical programming). We adopt a simple network as suggested by Das et al. (2017), which encodes the relational path using the recurrent neural network, computing the score using the similarity between path representation and relation representation. For each training sample $(r_\text{h}, (r_0, \ldots, r_i))$. Suppose $\hat{\boldsymbol{y}}$ denotes the output logits of the rule discriminator; the objective can be expressed as:

$$
\mathcal{L}_\text{Discriminator} = -\sum_{\mathcal{S}} \sum_r^{|\mathcal{R}|} \boldsymbol{y}_r^{r_\text{h}} \log \hat{\boldsymbol{y}}_r,
\tag{3}
$$

where $\mathcal{S}$ represents all the training samples and $\boldsymbol{y}^{r_\text{h}}$ is a one-hot vector that only the $r_\text{h}$-th position is 1.

## 2.3 Rule Generation

**Model Training** We train both the relational path autoencoder and the rule discriminator on the given knowledge graph $\mathcal{G}$. For the relational path autoencoder, we adopt a random walk (Spitzer, 1975)-based procedure to efficiently sample relational

paths to obtain training data. Each subsequent node is generated using the following distribution:

$$
p(x_i|x_{i-1}) = \begin{cases} \dfrac{1}{|\mathcal{N}(x_{i-1})|}, & (x_i, \cdot, x_{i-1}) \in \mathcal{T}, \\ 0, & \text{otherwise}, \end{cases}
\tag{4}
$$

where $\mathcal{N}(x_i)$ denotes all the neighborhoods of entity $x_i$. Then we optimize the auto-encoder by minimizing the Eq. 1. For the rule discriminator, we employ a similar sampling strategy. Each time we sample the next entity $x_i$, we include relation $r_\text{h}$, which directly connects $x_o$ and $x_i$, to create a training sample $(r_\text{h}, (r_0, \ldots, r_i))$. Suppose $\hat{\boldsymbol{y}}$ denotes the output logits of the rule discriminator, the objective can be expressed as follows:

$$
\mathcal{L}_\text{Discriminator} = -\sum_{\mathcal{S}} \sum_r^{|\mathcal{R}|} \boldsymbol{y}_r^{r_\text{h}} \log \hat{\boldsymbol{y}}_r,
\tag{5}
$$

where $\mathcal{S}$ represents all the training samples and $\boldsymbol{y}^{r_\text{h}}$ is a one-hot vector that only the $r_\text{h}$-th position is 1.

**Latent Vector Sampling** Given the joint distribution $p(\boldsymbol{z}, s)$, we would like to draw samples $\boldsymbol{z}$ conditioned on the target rule head $s$, which are then fed to the VAE decoder to obtain the desired rule bodies. According to Song et al. (2021), sampling from an EBM can be achieved by solving a specific ordinary differential equation (ODE). In our work, the ODE in the latent space can be expressed as: $d\boldsymbol{z} = \frac{1}{2}\beta(t)E_\theta(s|g(\boldsymbol{z}))dt$, with negative time increments from $T$ to $0$. To generate latent vectors based on the given rule head $s$, we first draw $\boldsymbol{z}(T)$ from $\mathcal{N}(\boldsymbol{0}, \boldsymbol{I})$, and then apply a neural ODE solver[1] (Chen et al., 2018, 2021) to

---

[1] https://github.com/rtqichen/torchdiffeq

| Methods | FB15k-237 | | | | WN18RR | | | |
| --- | --- | --- | --- | --- | --- | --- | --- | --- |
| | **MRR** | **Hits@1** | **Hits@3** | **Hits@10** | **MRR** | **Hits@1** | **Hits@3** | **Hits@10** |
| TransE | 0.294 | - | - | 46.5 | 0.226 | - | - | 50.1 |
| DistMult | 0.241 | 15.5 | 26.3 | 41.9 | 0.430 | 39.0 | 44.0 | 49.0 |
| TuckER | 0.358 | 26.6 | 39.4 | 54.4 | 0.470 | 44.3 | 48.2 | 52.6 |
| RotatE | 0.338 | 24.1 | 37.5 | 53.3 | 0.476 | 42.8 | 49.2 | 57.1 |
| PathRank[†] | 0.087 | 7.4 | 0.2 | 11.2 | 0.189 | 17.1 | 20.0 | 22.5 |
| NeuralLP[†] | 0.237 | 17.3 | 25.9 | 36.1 | 0.381 | 36.8 | 38.6 | 40.8 |
| DRUM[†] | 0.238 | 17.4 | 26.1 | 36.4 | 0.382 | 36.9 | 38.8 | 41.0 |
| RNNLogic[†] | 0.288 | 20.8 | 31.5 | 44.5 | 0.455 | 41.4 | 47.5 | 53.1 |
| RLogic[‡] | 0.312 | 20.3 | - | 50.1 | 0.473 | 44.3 | - | 53.7 |
| LATENTLOGIC | **0.320** | **21.2** | **32.9** | **51.4** | **0.481** | **45.2** | **49.7** | **55.3** |

Table 1: Knowledge graph reasoning performance on FB15k-237 and WN18RR. The model with [†] means the results are from Qu et al. (2021), and [‡] means the results are copied from Cheng et al. (2022). For RNNLogic, we choose the variant *w/o emb.* for a fair comparison.

obtain $z = z(0)$.

**Rule Generation** We draw $n$ samples $\{z_i(T)\}_{i=0}^n$ from $\mathcal{N}(0, I)$ for each rule head $s$ and obtain $n$ latent vectors $\{z_i(0)\}_{i=0}^n$. Then, we feed $\{z_i(0)\}_{i=0}^n$ to pretrained VAE decoder to generate corresponding rule bodies. Finally, we calculate the confidence scores of the generated rules using the aforementioned rule discriminator.

## 3 Experimental Setup

**Datasets** We conduct experiments on two widely used knowledge graph reasoning benchmark datasets: FB15k-237 (Toutanova and Chen, 2015) and WN18RR (Dettmers et al., 2018). WN18RR comprises of 40,943 entities and 11 relations, with 86k/3k/3k instances set aside for training/validation/testing correspondingly. FB15k-237 comprises 14,541 entities and 237 relations and has 272k/17k/20k instances reserved for training/validation/testing, respectively.

## 4 Experimental Results

**Baselines** We compared LATENTLOGIC with two taxonomies of methods, including: Knowledge graph embedding methods: TransE (Bordes et al., 2013), DistMult (Yang et al., 2015), TuckER (Balazevic et al., 2019), and RotatE (Sun et al., 2019). Rule-based methods: PathRank (Lee et al., 2013), NeuralLP (Yang et al., 2017), DRUM (Sadeghian et al., 2019), RNNLogic (Qu et al., 2021), and RLogic (Cheng et al., 2022).

**Evaluation Metrics** We adopt *forward chaining* (Salvat and Mugnier, 1996) for inferring missing facts from logical rules. For every test triplet $(h, r, t)$, two queries are created: $(h, r, ?)$ and $(?, r, t)$, using $t$ and $h$ as answers, respectively. To maintain consistency with previous studies, Mean Rank (MR), Mean Reciprocal Rank (MRR), and hit@k are selected as evaluation metrics under the *filtered* setting (Sun et al., 2019). Furthermore, to mitigate *the effects of random sampling*, we evaluate the model performance on five different random seeds, and report the average performance.

**Overall Performance** As shown in Tab. 1, we present the experimental results on the FB15k-237 and WN18RR datasets. Firstly, we compare LATENTLOGIC with rule-based methods and observe that LATENTLOGIC outperforms the statistical learning method PathRank, neural differentiable methods NeuralLP and DRUM, as well as recent methods RNNLogic and RLogic. In particular, we obtain 2.56% and 2.59% relative increase in MRR and Hits@10 on FB15k-237 against the state-of-the-art rule-based method RLogic (Cheng et al., 2022). Similarly, LATENTLOGIC achieves 1.69% and 2.98% increase in MRR and Hits@10 on the WN18RR dataset against RLogic. Then, we also compare LATENTLOGIC against salient knowledge graph embedding-based methods and find that LATENTLOGIC yields comparable performance to embedding-based methods, especially on WN18RR, where it outperforms selected embedding-based baselines.

**Quality of Learned Rules** We assess the quality of the logic rules learned from different models in this part. Following the settings of Qu et al. (2021),

| |
|---|
| $\texttt{film\_language}(x,y) \leftarrow \texttt{film\_actor}(x,z) \wedge \texttt{person\_languages}(z,y)$ 
 $\texttt{film\_language}(x,y) \leftarrow \texttt{film\_prequel}(x,z) \wedge \texttt{film\_language}(z,y)$ |
| $\texttt{film\_country}(x,y) \leftarrow \texttt{film\_produced\_by}(x,z) \wedge \texttt{person\_nationality}(z,y)$ 
 $\texttt{film\_country}(x,y) \leftarrow \texttt{film\_produced\_by}(x,z_1) \wedge \texttt{people\_lived\_location}(z_1,z_2) \wedge \texttt{location\_country}(z_2,y)$ |
| $\texttt{person\_nationality}(x,y) \leftarrow \texttt{place\_of\_birth}(x,z) \wedge \texttt{location\_country}(z,y)$ 
 $\texttt{person\_nationalit}(x,y) \leftarrow \texttt{organization\_founder}(x,z) \wedge \texttt{organization\_country}(z,y)$ |

Table 2: Case study on the FB15k-237 dataset.

we generate $n$ logic rules with the highest quality score for each query relation and then use them for training a predictor for knowledge graph reasoning. As mentioned before, the quality of each rule learned by LATENTLOGIC can be calculated using the rule discriminator. Results for different $n$ values are reported in Fig. 2. Our observations suggest that LATENTLOGIC outperforms the compared methods remarkably. Moreover, even with a limited number of rules considered per relation, LATENTLOGIC still achieves competitive results.

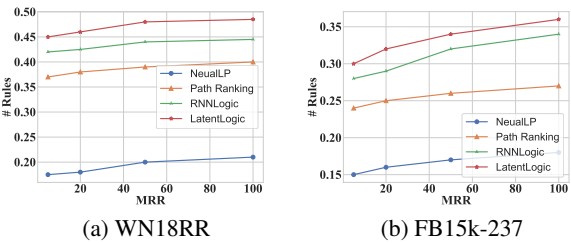

(a) WN18RR      (b) FB15k-237

Figure 2: MRR performance w.r.t. $n$ rules on WN18RR and FB15k-237 datasets. LATENTLOGIC achieves prominent results even with only 10 rules.

**Case Study**     To demonstrate that LATENTLOGIC is capable of generating helpful and varied rules for knowledge graph reasoning, we present some of the logic rules that LATENTLOGIC produced on the FB15k-237 dataset in Tab. 2. It can be observed that the logic rules generated by LATENTLOGIC are semantically meaningful and diverse.

## 5   Related Work

Over the past years, learning logical rules over knowledge graphs has been an active research area. Most traditional methods enumerate relational paths between query entities and answer entities as candidate logic rules, and further learn a scalar weight for each rule to assess the quality. Some representative works include Markov Logic Network (MLN) (Kok and Domingos, 2005; Richardson and Domingos, 2006), path ranking (Lao and Cohen, 2010; Lao et al., 2011) and probabilistic personalized page rank (ProPPR) algorithms (Wang et al.,

2013, 2014a,b). Then some methods extend the idea by simultaneously learning logic rules and the weights in a differentiable way. For example, some works (Rocktäschel and Riedel, 2017; Yang et al., 2017; Sadeghian et al., 2019) try to use neural logic programming to model rule-based reasoning and to learn high-quality logical rules. Another kind of rule-learning method is based on reinforcement learning. The basic idea is to train a path-finding agent, which is used to search for reasoning paths over knowledge graphs to answer queries, and then extract logic rules from reasoning paths. Some representative works include Deep-Path (Xiong et al., 2017b), MINERVA (Das et al., 2018), M-Walk (Shen et al., 2018) and R5 (Lu et al., 2022). Recently, RNNLogic (Qu et al., 2021) solves this problem by training a generator and a predictor alternately. RLgoic (Cheng et al., 2022) learns rules in a recursive manner.

## 6   Conclusion

Learning logic rules for knowledge graph reasoning is crucial, as they offer interpretable explanations for the process of reasoning. We propose a novel framework for learning logic rules in latent space. Concretely, we introduce a relational path autoencoder to map paths into latent space and a rule discriminator to access the consistency between rule bodies and rule heads. With the sampler based on ODE, LATENTLOGIC can generate logic rules efficiently. Experimental results showed that LATENTLOGIC outperforms strong baseline methods and is efficient for training.

## Acknowledgements

We thank the anonymous reviewers for their insightful feedback. This work is supported by the National Natural Science Foundation of China (No.U20B2053). We thank the Beijing Advanced Innovation Center for Big Data and Brain Computing for its computational resources.

## Limitations

In this paper, we introduce a generative framework for rule mining on knowledge graphs. We believe this approach still has much room for improvement: 1) We have not used the more complicated model architecture of the rule discriminator. 2) The proposed sampling strategy may lead to some same relational paths or out-of-domain relational paths. For future work, finding a way to incorporate a more advanced rule discriminator and prevent generating invalid relational paths is worth exploring.

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

# 7  Appendix

## 7.1  Dataset Statistics

In this part, we provide a brief introduction and statistics of FB15k-237 and WN18RR, which are used in this paper.

- FB15k-237 is one of the most commonly used benchmark knowledge graph datasets, which is an online collection of structured data harvested from many sources, including individual, user-submitted wiki contributions.

- WN18RR is recognized as a widely adopted benchmark knowledge graph dataset. Its purpose is to generate a user-friendly dictionary and thesaurus and facilitate automatic text analysis. The entities in the dataset represent word senses, while relationships define the lexical relations between these senses.

| Dataset | # Data | # Relation | # Entity |
|---------|--------|------------|----------|
| FB15k-237 | 310,116 | 237 | 14,541 |
| WN18RR | 93,003 | 11 | 40,943 |

Table 3: Dataset statistics.

## 7.2  Implementation Details

We implement LATENTLOGIC over Pytorch[2]. We use the Adam optimizer to train both the relational path autoencoder and the rule discriminator. A grid search is performed to determine the best hyperparameters based on the performance on the validation sets. We employ the *dopri5* neural ODE solver, with $(10^{-3}, 10^{-3})$ tolerances, and set $T = 1$, $\beta_{\min} = 0.1$, and $\beta_{\max} = 20$. The maximum length of relational paths is set to 3 in FB15k-237 and 2 in WN18RR. All experiments are executed on a single Nvidia Tesla V100 GPU.

## 7.3  Details of Forward Chain

We provide the details of how to use *forward chain* to infer missing facts given learned logic rules. Specifically, Given a query $(h, r, t)$, let $\mathcal{A}$ be the set of candidate answers that can be discovered by any learned rule using forward chaining. For each candidate answer $a \in \mathcal{A}$, the score of triple $(h, r, a)$ can be calculated as $\sum_{rule} \sum_{path} \phi(rule)$, where $\phi$ is the confidence score of logic rule. Then we can rank candidate answers by the scores.

[2] https://pytorch.org/

| Methods | FB15k-237↓ (mins) | WN18RR↓ (mins) |
|---------|-------------------|----------------|
| NerualLP | 395 | 122.3 |
| DRUM | 373.8 | 118.7 |
| RNNLogic | 331.6 | 100.3 |
| LATENTLOGIC | 24.6 | 12.4 |

Table 4: Training time running on a single NVIDIA Tesla V100 GPU.

## 7.4  Supplementary Experiment

**Evaluation of Training Efficiency**  To showcase the efficiency of LATENTLOGIC, we compare the training time of different models in Tab. 4. For a fair comparison, we utilize a single GPU (Tesla V100) to train each model and implement the hyperparameters recommended by the original papers. Our observations are as follows: (i) LATENTLOGIC outperforms the baselines in efficiency while still achieving competitive performance. (ii) NeuralLP and DRUM do not perform well due to their involvement in large matrix multiplication. (iii) RNNLogic is also less efficient because of the EM-based optimization procedure.

**Sensitivity of Randomness**  Since our approach relies on random sampling techniques, we would like to conduct experiments to examine the sensitivity of LATENTLOGIC to randomness. In Tab.5, we report the mean and standard deviations of the results under different random seeds. We can notice that the randomness sampling has minimal impact on the performance of LATENTLOGIC.

| | FB15k-237 | | | WN18RR | | |
|---|-----|------|------|-----|------|------|
| | MRR | H@1 | H@10 | MRR | H@1 | H@10 |
| $\mu$ | 0.320 | 21.2 | 51.4 | 0.481 | 45.2 | 55.3 |
| $\sigma$ | 0.013 | 0.118 | 0.247 | 0.009 | 0.211 | 0.187 |

Table 5: result w.r.t. randomness.