# OpenReview forum: "LATENTLOGIC: Learning Logic Rules in Latent Space over Knowledge Graphs"
_EMNLP/2023/Conference — EMNLP 2023 Findings_

### Official Review · Reviewer_GJYL · 2023-08-03

**Soundness:** 3

**Excitement:**

3: Ambivalent: It has merits (e.g., it reports state-of-the-art results, the idea is nice), but there are key weaknesses (e.g., it describes incremental work), and it can significantly benefit from another round of revision. However, I won't object to accepting it if my co-reviewers champion it.

**Missing References:**

[a] A Survey on Knowledge Graphs: Representation, Acquisition, and Applications, https://ieeexplore.ieee.org/abstract/document/9416312
[b] Low-Dimensional Hyperbolic Knowledge Graph Embeddings, ACL.

**Paper Topic And Main Contributions:**

In this paper, the authors proposed a knowledge graph completion method, LATENTLOGIC.  It employs a pre-trained VAE to map relational paths into latent space, and uses a sampler based ODE to enable efficient generation of logic rules, aiming to avoid the inefficient optimization issue. To verify the effectiveness, they conducted experiments on two knowledge graph datasets and show that the proposed approach can outperform a number of embedding-based/rule-mining based approaches.

**Questions For The Authors:**

1. It would have been appreciated to compare qualitative results of the different methods.
2. What is training time like of embedding based methods?
3. What is inference time like?
4. Are all the learned rules intuitively reasonable?



**Reasons To Accept:**

1. The proposed solution is novel. Specifically, the proposed framework utilizes fancy component such as energy-based models and ordinary differential equations to boost the efficiency of effectiveness, which is novel and interesting.
2. The training time can be greatly reduced compared with other rule-based methods (as shown in Table 2).
3. Qualitative analyses further justify the effectiveness of the rule-mining process.
4. The presentation is clear and easy to follow.

**Reasons To Reject:**

1. The performance is not state-of-the-art. Only a limited number of methods are compared in table 1. Check [a] for more methods.
2. Only two datasets are used, more datasets such as Kinship, UMLS, YAGO3, etc. can be used.

[a] A Survey on Knowledge Graphs: Representation, Acquisition, and Applications, https://ieeexplore.ieee.org/abstract/document/9416312

**Reproducibility:**

4: Could mostly reproduce the results, but there may be some variation because of sample variance or minor variations in their interpretation of the protocol or method.

**Reviewer Confidence:**

4: Quite sure. I tried to check the important points carefully. It's unlikely, though conceivable, that I missed something that should affect my ratings.

---

> ### Author Rebuttal · Authors · 2023-08-27
>
> Dear Reviewer GJYL,
>
> We sincerely appreciate the valuable feedback provided by the reviewer.  We now address the concerns raised by the reviewer one by one as below.
>
> > Q1: The performance is not state-of-the-art. Only a limited number of methods are compared in table 1. Check [a] for more methods.Only two datasets are used, more datasets such as Kinship, UMLS, YAGO3, etc. can be used.
>
> Thank you for your suggestion, and we are sorry that we did not include more baselines and data sets due to space limitations, and we choose FB15k-237 and WN18RR since they are the most commonly used benchmark datasets.
> We provide experimental results on UMLS below. We will try to add experimental results on additional data sets in the final version.
>
> | Methods | MRR | Hits@11 |
> | --- | --- | --- |
> | RotatE | 0.925 | 86.3 |
> | DRUM | 0.813 | 67.4 |
> | RNNLogic | 0.842 | 77.2 |
> | LATENTLOGIC | 0.859 | 81.7 |
>
> > Q2: It would have been appreciated to compare qualitative results of the different methods.
>
> Thank you for your suggestion. Taking the results from the FB15k-237 dataset as an example.
>
> | Methods | MRR | Hits@3 |
> | --- | --- | --- |
> | TransE | 0.226 | - |
> | TuckER | 0.470 | 48.2 |
> | RotatE | 0.476 | 49.2 |
> | DRUM | 0.382 | 38.8 |
> | RNNLogic | 0.455 | 47.5 |
> | LATENTLOGIC | 0.481 | 49.7 |
>
> We can observe that the  LATENTLOGIC performs better than the salient rule-based method (RNNLogic) with a relative improvement of 5.7% in MRR and 4.6% in Hits@3. Furthermore, LATENTLOGIC also outperforms the prominent embedding-based method (RotatE) with a relative improvement of 1.1% in MRR.
>
> > Q3: What is training time like of embedding based methods?
>
> Thank you for your question.  We report the training time of embedding-based methods on FB15k-237 dataset. The hyperparameters used are identical to those specified in [a]. All experiments were conducted on a single NVIDIA Tesla V100 GPU, consistent with the paper.
>
> [a] Sun, Z., Deng, Z. H., Nie, J. Y., & Tang, J. (2019). Rotate: Knowledge graph embedding by relational rotation in complex space.
>
> |  | FB15k-237(mins) |
> | --- | --- |
> | TransE | 51.3 |
> | DistMult | 83.5 |
> | RotatE | 165.6 |
>
> > Q4: What is inference time like?
>
> The table below presents the inference time on the test sets for our method. The test set for FB15k-237 consists of 40,932 triplets, while the test set for WN18RR comprises 6,268 triplets.
>
> |  | FB15k-237(mins) | WN18RR(mins) |
> | --- | --- | --- |
> | LATENTLOGIC | 11.13 | 3.77 |
>
> > Q5: Are all the learned rules intuitively reasonable?
>
> Thank you for your question. We can find that most of the rules are reasonable. Although some rules on some datasets (wn18rr) cannot be described in natural language, they still logically established.
>
> ---
>
> Finally,  Thank you for all the constructive feedback and positive comments. We will add the aforementioned additional details you   concern or suggested in the final version.
>
> Sincerely & Best wishes

---

### Official Review · Reviewer_rqqL · 2023-08-05

**Soundness:** 4

**Excitement:**

3: Ambivalent: It has merits (e.g., it reports state-of-the-art results, the idea is nice), but there are key weaknesses (e.g., it describes incremental work), and it can significantly benefit from another round of revision. However, I won't object to accepting it if my co-reviewers champion it.

**Paper Topic And Main Contributions:**

This paper proposes a novel approach for extracting rules from Knowledge bases. Their approach relies on creating a latent space where they can use latent vectors to draw samples for rule heads and then use a pretrained VAE decoder to generate rule bodies for each sampled head. The quality of the rules is verified by an energy-based discriminator. They provide results on two datasets and a series of different methods based on embedding and rules. They show their approach can mainly surpass the prior rule-based methods.

**Questions For The Authors:**

- What is the VAE autoencoder that you use?! You mention this is pretrained.
- Could you describe in more detail as to how the rule discriminator scores the extracted rules?!
- It is unclear how you measure the quality of the extracted rules. Could you elaborate on that?!


**Reasons To Accept:**

- The method is novel and better than prior research for rule-based completion of knowledge graphs.
- The paper showcases a proper comparison with a long list of baselines.

**Reasons To Reject:**

- There is a need to explain why the method fails in getting better results than some of the embedding-based methods, especially on the FB15k dataset.
- The implementation details are missing.
- The paper is missing some details which would have been better explained in a long paper rather than the current short version.


**Reproducibility:**

3: Could reproduce the results with some difficulty. The settings of parameters are underspecified or subjectively determined; the training/evaluation data are not widely available.

**Reviewer Confidence:**

3: Pretty sure, but there's a chance I missed something. Although I have a good feel for this area in general, I did not carefully check the paper's details, e.g., the math, experimental design, or novelty.

**Typos Grammar Style And Presentation Improvements:**

- Line 245: Why is the font different for "the effect of random sampling"?!
- Line 082: Please write the full name of VAE in its first use in the paper.

---

> ### Author Rebuttal · Authors · 2023-08-27
>
> Dear Reviewer rqqL,
>
> We thank the reviewer for all the insightful comments. We will add more technical details and implementation details in the final version and fix the typos and errors.
> In the following sections, we address each concern raised by the reviewer individually.
>
> > Q1: There is a need to explain why the method fails in getting better results than some of the embedding-based methods, especially on the FB15k dataset.
>
> Thank you！This is an intriguing question. It is commonly observed that rule-based methods tend to underperform in comparison to embedding-based methods. We attribute this discrepancy to the incompleteness of the knowledge graph. To illustrate, let us consider the rule $\texttt{Nationality} \wedge \texttt{OfficialLanguage} \rightarrow \texttt{SpeakLanguage}$. With this rule, we can deduce that $\texttt{Nationality}(Jordan, USA) \wedge \texttt{OfficialLanguage}(USA,English)$ implies $\texttt{SpeakLanguage} (Jordan, English)$. However, if the fact $\texttt{Nationality}(Jordan, USA)$ is missing from the background knowledge graph, rule-based methods cannot accurately predict this fact.
>
> > Q2: What is the VAE autoencoder that you use?! You mention this is pretrained.
>
> Thank you for your question！In our approach, we utilize a Variational Autoencoder (VAE) with a Recurrent Neural Network (RNN)-based architecture. Specifically, we employ an RNN-based encoder to transform discrete relational paths into a continuous latent space, while applying an RNN-based decoder to reconstruct relational paths from a latent vector.
> In the training procedure, we collect training data by sampling relational paths from the knowledge graph (KG).
>
> > Q3: Could you describe in more detail as to how the rule discriminator scores the extracted rules?!
>
> Thank you for your question！We assign an embedding vector to each relation. The rule discriminator utilizes an RNN to encode the relation embeddings of the relational path into a path representation. It then generates a similarity score by comparing the path representation with the rule head relation representation.
> The rule discriminator is trained to predict the consistency score between relational paths and the rule head.
> For each extracted rule, the rule discriminator takes the rule head, denoted as$r_h$and rule body, represented as$(r_1, \ldots, r_n)$and outputs the consistency score between them.
>
> > Q4: It is unclear how you measure the quality of the extracted rules. Could you elaborate on that?!
>
> Thank you for your question！The quality of the extracted rules can be evaluated using the rule discriminator, as described in the previous question (Q3).
>
> ---
>
> We appreciate the reviewer's endorsement and constructive suggestions, which will enhance the final paper version.
>
> Sincerely and Best regards

---

### Official Review · Reviewer_dB1w · 2023-08-13

**Soundness:** 3

**Excitement:**

4: Strong: This paper deepens the understanding of some phenomenon or lowers the barriers to an existing research direction.

**Paper Topic And Main Contributions:**

The paper delves into the realm of learning logic rules over knowledge graphs. Traditional enumeration methods grapple with an expansive search space, while Reinforcement Learning and EM methodologies are hindered by intricate optimization processes. Addressing these challenges, this paper introduces a method of controllable sampling in latent space, which effectively sidesteps the need to enumerate relational paths. Central to this approach is a VAE-based autoencoder, which projects the relational paths into a continuous latent space. A discriminator is then leveraged to gauge the semantic coherence between the latent vector and the rule head. Further, a neural ODE-based sampler extracts latent vector samples corresponding to the targeted rule head. Comprehensive experiments corroborate the method's enhanced scalability and its capacity to generate high-quality logic rules.

**Reasons To Accept:**

1. The paper is eloquently structured, with each component meticulously illustrated.
2. From a technical standpoint, the proposed method stands robust, underpinned by thorough experiments. Notably, it not only improves the quality of generated rules but also scales more effectively.

**Reasons To Reject:**

The study of learning logic rules over knowledge graphs, while valuable, has become somewhat saturated. Numerous methodologies have been put forth in recent times, and the incremental advancement presented by this paper seems relatively modest.

**Reproducibility:**

4: Could mostly reproduce the results, but there may be some variation because of sample variance or minor variations in their interpretation of the protocol or method.

**Reviewer Confidence:**

4: Quite sure. I tried to check the important points carefully. It's unlikely, though conceivable, that I missed something that should affect my ratings.

---

> ### Author Rebuttal · Authors · 2023-08-27
>
> Dear Reviewer dB1w,
>
> We would like to thank you for all the meticulous and constructive feedback and positive comments. We have elaborated on our responses as follows:
>
> - Based on our research and to the best of our knowledge, our work is the first to propose the concept of controllable generation for addressing rule learning problems.
>
> - This controllable generation approach effectively tackles the shortages of excessive search space and inefficient optimization, resulting in more efficient and effective rule learning methods compared to previous approaches such as Neural Logic Programming and EM-based methods like RNNLogic. We firmly believe that our work will serve as an inspiration for future research endeavors in this field, particularly in exploring the representation of latent space and modeling of joint distributions. In addition, this paradigm can be applied to other tasks involving discrete rule learning.
>
> ---
> Thanks again for your appreciation and constructive comments on our work!
>
> Sincerely & Best regards

---

### Official Review · Reviewer_Tpzv · 2023-08-21

**Soundness:** 3

**Ethical Concerns:**

Yes

**Excitement:**

3: Ambivalent: It has merits (e.g., it reports state-of-the-art results, the idea is nice), but there are key weaknesses (e.g., it describes incremental work), and it can significantly benefit from another round of revision. However, I won't object to accepting it if my co-reviewers champion it.

**Paper Topic And Main Contributions:**

This paper proposes a new framework called LATENTLOGIC for efficiently learning logic rules from knowledge graphs. The key ideas and contributions are:
Problem Addressed:
1.Learning interpretable and generalizable logic rules from knowledge graphs is important for reasoning and inference, but existing methods have challenges like huge search spaces and inefficient optimization.
Main Contributions:
1.Maps discrete relational paths to a continuous latent space using a VAE-based auto-encoder, avoiding enumerating all paths.
2.Introduces a rule discriminator on the latent space to measure consistency between rule body and head, forming an energy-based distribution for sampling.
3.Employs a neural ODE-based sampler for controlled generation of rule bodies given a rule head.
4.Achieves state-of-the-art results on FB15k-237 and WN18RR datasets compared to existing rule learning methods

**Reasons To Accept:**

Strengths:
1.Proposes a creative way of mapping the discrete relational path search problem into continuous latent space for more efficient optimization.
2.The model components (VAE, discriminator, ODE sampler) are nicely integrated for controlled logic rule generation.
3.Achieves new state-of-the-art results compared to prior logic rule learning methods.
4.Demonstrates improved training efficiency and scalability over existing approaches.
5.Provides interpretable logic rules that can explain predictions and generalize to new facts.

**Reasons To Reject:**

1.The latent space mapping idea is creative but also complex, making the model behavior somewhat harder to interpret. Additional visualizations or analyses could help provide insight.
2.It is not clear how the approach handles very long relational paths or complex rules. Testing the scalability on more challenging tasks could be beneficial.
3.The space of related methods compared against is limited. Comparing to a few additional recent techniques could better situate the contributions.
4.Certain details are lacking on hyperparameter tuning, optimal settings, and other implementation choices to ensure reproducibility.

**Reproducibility:**

3: Could reproduce the results with some difficulty. The settings of parameters are underspecified or subjectively determined; the training/evaluation data are not widely available.

**Reviewer Confidence:**

4: Quite sure. I tried to check the important points carefully. It's unlikely, though conceivable, that I missed something that should affect my ratings.

---

> ### Author Rebuttal · Authors · 2023-08-27
>
> Reviewer Tpzv,
>
> We express our gratitude to the reviewer for providing us with insightful comments. In the following, we address each concern raised by the reviewer individually:
>
> > Q1.The latent space mapping idea is creative but also complex, making the model behavior somewhat harder to interpret. Additional visualizations or analyses could help provide insight.
>
> We apologize for any confusion caused. We acknowledge that, due to space constraints in the current version, there is a lack of detailed description of our model and implementation. However, we will ensure to provide additional pages in the final version to include more comprehensive information on our model and implementation details. These additions will include an extended framework diagram to visualize more details, such as the rule discriminator.
> Thank you for bringing this to our attention.
>
> > Q2: It is not clear how the approach handles very long relational paths or complex rules. Testing the scalability on more challenging tasks could be beneficial.
>
> Since our model is built upon a generative model, it has the capability to handle longer relational paths. Nevertheless, based on previous research and our own experiments, we have observed that relational paths with a length greater than 5 have minimal impact on the reasoning process. This finding suggests that longer relation paths are often inferred from shorter ones. Therefore, we believe that there is limited beneficial information gained from excessively long relation paths in our model.
>
> > Q3: The space of related methods compared against is limited. Comparing to a few additional recent techniques could better situate the contributions.
>
> We acknowledge and appreciate the valuable advice, we have not included all related work due to space constraints. But we will try to add more recent work to highlight the contribution of our work in the final version.
>
> > Q4: Certain details are lacking on hyperparameter tuning, optimal settings, and other implementation choices to ensure reproducibility.
>
> ---
> Thank you for your valuable suggestions. In light of these suggestions, we will include the details of hyperparameter tuning and other implementation aspects in the appendix.
>
> Sincerely & Best wishes

---

### Meta-Review · Area_Chair_Nv4V · 2023-09-15

**Recommendation:** 2

**Metareview:**

This paper delves into the realm of learning logic rules on knowledge graphs for reasoning and introduces the LATENTLOGIC model, designed to mine logic rules through controllable generation within the latent space.

The overall consensus among most PC members is that this paper demonstrates a good soundness but elicits mixed feelings in terms of excitement. Several noteworthy weaknesses have been highlighted by the reviewers, including the high complexity of the proposed model, the absence of the latest baselines for comparison, insufficient details concerning the implementation, and potential limitations associated with the choice of datasets.

Considering the good soundness and the identified concerns, this paper has the potential to be accepted as a findings paper. However, the authors are encouraged to diligently address all the concerns outlined in the reviews during their new revision, in order to enhance the overall quality and impact of this paper.

---

### Decision · Program_Chairs · 2023-10-07

**Decision:**

Accept-Findings

**Comment:**

This paper delves into the realm of learning logic rules on knowledge graphs for reasoning and introduces the LATENTLOGIC model, designed to mine logic rules through controllable generation within the latent space.

The overall consensus among most PC members is that this paper demonstrates a good soundness but elicits mixed feelings in terms of excitement. Several noteworthy weaknesses have been highlighted by the reviewers, including the high complexity of the proposed model, the absence of the latest baselines for comparison, insufficient details concerning the implementation, and potential limitations associated with the choice of datasets.

Considering the good soundness and the identified concerns, this paper has the potential to be accepted as a findings paper. However, the authors are encouraged to diligently address all the concerns outlined in the reviews during their new revision, in order to enhance the overall quality and impact of this paper.